# The Immunogenicity and Properties of a Whole-Cell ETEC Vaccine Inactivated with Psoralen and UVA Light in Comparison to Formalin

**DOI:** 10.3390/microorganisms11082040

**Published:** 2023-08-09

**Authors:** Marlena M. Westcott, Maria Blevins, Thomas F. Wierzba, Alexis E. Morse, Kinnede R. White, Leigh Ann Sanders, John W. Sanders

**Affiliations:** 1Department of Microbiology and Immunology, Wake Forest University School of Medicine, 575 Patterson Ave, Winston Salem, NC 27101, USA; aemorse@wakehealth.edu (A.E.M.); krwhit7@emory.edu (K.R.W.); 2Infectious Diseases Section, Wake Forest University School of Medicine, Medical Center Blvd, Winston Salem, NC 27157, USA; mblevins@wakehealth.edu (M.B.); twierzba@wakehealth.edu (T.F.W.); lasander@wakehealth.edu (L.A.S.); jwsander@wakehealth.edu (J.W.S.)

**Keywords:** ETEC, psoralen, formalin, vaccine, dmLT, mice

## Abstract

Inactivated whole-cell vaccines present a full repertoire of antigens to the immune system. Formalin treatment, a standard method for microbial inactivation, can modify or destroy protein antigenic epitopes. We tested the hypothesis that photochemical inactivation with psoralen and UVA light (PUVA), which targets nucleic acid, would improve the immunogenicity of an Enterotoxigenic *E. coli* (ETEC) vaccine relative to a formalin-inactivated counterpart. Exposure of ETEC H10407 to PUVA using the psoralen drug 4′-Aminomethyltrioxsalen hydrochloride (AMT) yielded replication-incompetent bacteria that retained their metabolic activity. CFA/I-mediated mannose-resistant hemagglutination (MRHA) was equivalent for PUVA-inactivated and live ETEC, but was severely reduced for formalin–ETEC, indicating that PUVA preserved fimbrial protein functional integrity. The immunogenicity of PUVA–ETEC and formalin–ETEC was compared in mice ± double mutant heat-labile enterotoxin (dmLT) adjuvant. Two weeks after an intramuscular prime/boost, serum anti-ETEC IgG titers were similar for the two vaccines and were increased by dmLT. However, the IgG responses raised against several conserved ETEC proteins were greater after vaccination with PUVA–ETEC. In addition, PUVA–ETEC generated IgG specific for heat-labile toxin (LT) in the absence of dmLT, which was not a property of formalin–ETEC. These data are consistent with PUVA preserving ETEC protein antigens in their native-like form and justify the further testing of PUVA as a vaccine platform for ETEC using murine challenge models.

## 1. Introduction

New vaccine approaches that are safe and effective for widespread use, including in individuals with compromised immune systems, are needed. Killed whole-cell vaccines have advantages over protein subunit, nucleic acid-based, and viral vectored vaccines in that they present a broad repertoire of strain-specific and conserved protein and polysaccharide antigens to the immune system and are generally safe for use in vulnerable individuals. Killed whole-cell vaccines are agile in that they require minimal or no molecular engineering or preexisting knowledge of the role of specific proteins in inducing protective immune responses. They are relatively inexpensive to produce and are compatible with multivalency.

Formalin treatment has served as a standard method for killing microorganisms for the purpose of producing vaccines. Formalin modifies amino acids by the addition of reactive carbonyl groups, promoting protein crosslinks. Although formalin inactivation has produced effective vaccines, for example, the Hepatitis A vaccine (HAV) [1], detrimental effects on protein antigenic integrity, immunogenicity, and/or the production of functional antibodies have been demonstrated for vaccines, including *Shigella* [2], *Pseudomonas aeruginosa* [3], *Bordetella pertussis* [4], *Salmonella* [5] poliovirus [6,7], respiratory syncytial virus [8], and others [9]. Because formalin has the potential to negatively impact proteins in the context of at least some antigenic epitopes, alternative inactivation methods that preserve protein antigenic structure could improve the immunogenicity of killed vaccines.

Treatment with psoralen drugs followed by irradiation with long-wavelength UV light (UVA, 315–400 nm) disables the replication of viruses [10,11] and bacteria [12] by introducing intra- and inter-strand crosslinks into nucleic acid at pyrimidine residues [13]. Photochemical inactivation with psoralen drugs effectively inactivates multiple families of RNA viruses while preserving viral protein antigenic integrity [10,14]. Virus inactivation with the psoralen drug 4′-Aminomethyltrioxsalen hydrochloride (AMT) generated Dengue virus vaccines that were highly immunogenic in mice [15,16] and nonhuman primates (NHP) [17]. Importantly, AMT-killed Dengue virus retained monoclonal antibody reactivity against key protein antigenic epitopes and promoted T cell responses that were similar to those generated by the live virus, while both properties were diminished with the formalin-killed virus [16]. The superiority of psoralen over formalin inactivation was demonstrated for an AMT psoralen-inactivated tetravalent Dengue virus vaccine, as measured by the neutralizing antibody titers in mice and NHPs [18]. 

In the context of bacterial vaccines, psoralen + UVA has yielded promising results in preclinical studies for multiple pathogens [12]. The treatment of bacteria with the psoralen drugs amotosalen (S59) and 8-MOP disabled replication and conferred a transient state of metabolic activity (killed but metabolically active, KBMA) by virtue of sustained transcription in the chromosomal regions in between crosslinks [12,19]. Psoralen-killed KBMA bacteria produced and secreted endogenous or heterologous protein antigens capable of stimulating humoral and cellular immune responses when administered to mice, as demonstrated for *Salmonella typhimurium* [20] *Pseudomonas aeruginosa* [21,22], *Listeria monocytogenes* [19], and *Bacillus anthracis* [23]. In that respect, psoralen-inactivated bacteria resemble live, attenuated bacteria, and it has been proposed that the KBMA property could be leveraged to improve whole-cell bacterial vaccines and bacterial vaccine vectors [12]. Despite evidence that psoralen inactivation confers properties that could improve bacterial vaccines, no psoralen-killed bacterial vaccine candidate has advanced beyond preclinical studies. In addition, studies comparing psoralen to formalin-inactivated bacterial vaccines are lacking, leaving the formalin standard largely unchallenged. 

The current study was undertaken to compare psoralen inactivation to formalin inactivation using Enterotoxigenic *E. coli* (ETEC), for which psoralen inactivation for the purpose of vaccine development has not previously been reported. ETEC is a significant cause of diarrheal illness in the developing world, predominantly affecting children under the age of 5 [24,25]. In addition to causing significant childhood morbidity, acute diarrheal episodes are associated with growth stunting, developmental delays, and malnutrition [26]. To date, ETEC vaccine efforts have primarily focused on surface protein adhesins and the heat-labile enterotoxin (LT), which give rise to protective immune responses [27]. Adhesion to small intestinal epithelial cells, an essential step in ETEC pathogenesis, is mediated largely by colonization factors (CF) and coli surface proteins (CS) [28]. Consideration of the relative distribution of surface adhesins among ETEC clinical isolates has led to a focus on CFA/I, CS3, CS5, and CS6 as targets for a broad-coverage vaccine and to the development of ETVAX, an oral vaccine comprising killed, nonpathogenic *E. coli* that overexpress CFA/I, CS3, CS5, and CS6. ETVAX is delivered with LCTBA toxoid and the adjuvant dmLT [29]. DmLT (LT(R192G/L211A) is a detoxified derivative of LT [30] that retains the inherent adjuvant activity of the native toxin, including the ability to promote Th17 and IgA-producing B cells at mucosal surfaces after co-administration with a variety of antigens via multiple routes [31]. In addition to ETVAX clinical trials [32,33], dmLT has been tested in other human trials, for example to adjuvant an inactivated poliovirus vaccine [34]. Efforts to identify conserved and protective protein antigens that could extend the range of strains covered by ETEC vaccines have revealed new targets [35,36,37,38]. A method of inactivation that leaves surface proteins preserved in a configuration that resembles live bacteria could maximally leverage conserved proteins to enhance protective immunity.

Here, we established experimental conditions for the photochemical inactivation of ETEC using the psoralen drug 4′-Aminomethyltrioxsalen hydrochloride (AMT), which has previously been used to inactivate viruses [10,15]. The goals of the study were to compare the in vitro properties of psoralen-inactivated ETEC (PUVA–ETEC) to formalin-inactivated ETEC (formalin–ETEC) and to measure their relative immunogenicity in mice. We also assessed the value of adding dmLT as an adjuvant using an intramuscular (IM) prime-boost vaccination model. The results indicated that psoralen-inactivated bacteria have unique properties that could improve ETEC vaccine immunogenicity and, as such, serve as a foundation for efficacy studies against existing vaccines in ETEC infection models. 

## 2. Materials and Methods

Bacterial strains and growth conditions. The ETEC strain H10407 (O78:H11), expressing LT, ST, and CFA/I [39], was acquired from ATCC (#35401). ETEC B7a (O148:H28) [40], expressing LT, ST, and CS6, was acquired from the Naval Medical Research Center, Bethesda, MD, USA. The bacteria were maintained on CFA or LB agar, with fresh cultures started weekly from frozen stocks (25% glycerol in LB broth, −80 °C). For experiments, ETEC was cultured in CFA or LB broth, as indicated.

Psoralen inactivation. Overnight CFA broth cultures (37 °C with shaking) were diluted 1:500 into 1.0 mL of fresh media and, after 5 h of culture bacteria (OD_600_ 1.5), were treated with AMT psoralen (4′-Aminomethyltrioxsalen hydrochloride, #A4330, Sigma –Aldrich, Inc., St. Louis, MO, USA). For dose–response experiments, the range was 0.1–100 µg/mL; 0.34–340 µM. After an additional 1 h at 37 °C with shaking, 1.0 mL cultures were transferred to individual wells of a 6-well tissue culture dish and exposed (lid on to maintain sterility) to the indicated dose of UVA light at 365 nm (dose range 1.0–4.0 J/cm^2^) using an Analytik Jena Crosslinker (model# CL-1000L, Upland, CA, USA). The treated bacteria were recovered from the wells and washed 3 times before plating the dilutions onto LB agar to enumerate colony forming units (CFU). Bacteria treated with AMT only and UVA light only served as controls. To detect residual CFU, the entire 1.0 mL culture was plated on LB agar. As an additional read-out of residual replication-competent bacteria, 1.0 mL of inactivated or live bacteria was inoculated into 50 mL of fresh broth and monitored for growth at 37 °C over a period of 72 h. 

Formalin inactivation. ETEC were formalin-inactivated using the method of Tobias et al. [41]. The bacteria cultured as per the psoralen inactivation procedure were harvested at 5 h, washed, adjusted to 1 × 10^10^ CFU/mL in PBS, and treated with 1.5% formalin (0.2 M) for 2 h with shaking at 37 °C. The bacteria were then moved to 4 °C and incubated static for 3 days. The treated bacteria were washed 3 times and killing was confirmed by plating on LB agar. 

Killed but metabolically active (KBMA) assay. Metabolic activity was assessed by MTS assay using a CellTiter 96 AQ_ueous_ nonradioactive Cell Proliferation Assay (Promega (Chuo City, Tokyo)), which measures the activity of cellular dehydrogenase enzymes. After PUVA or formalin inactivation, the bacteria were washed, resuspended in LB broth, and added to 96-well plates (25 µL of bacteria in 75 µL of LB) + 10 µL/well MTS reagent. The controls were live and heat-killed bacteria and growth media only. The cultures were incubated for 1 h at 37 °C, after which, the absorbance of the soluble formazan product was measured at 490 nm using a BioTek EPOH2NS microplate reader with Gen5 software, version 3.1. 

MRHA assay. The capacity of CFA/I fimbriae to mediate the mannose-resistant agglutination of red blood cells [42] was performed as a measure of CFA/I structural/functional integrity after inactivation. ETEC H10407 was grown in static CFA broth cultures at 37 °C for 24 h, then diluted 1:100 into fresh cultures. After an additional 24 h static culture period, bacteria were inactivated. For PUVA inactivation, AMT (50 µg/mL) was added for the last 1.5 h of culture with periodic mixing, and then irradiation was performed as described above. Formalin inactivation of static-grown bacteria was performed as described for shaking cultures. After inactivation, the bacteria were washed and adjusted to OD_600_ = 1.0. The MRHA assay was performed in 96-well round bottom plates and on glass slides. For the 96-well plate assay, 100 µL of bacteria was mixed with 50 µL of PBS + 2% mannose or PBS only. Then, 50 µL/well of washed, Type A human RBC (ZenBio, Cary, NC, USA) adjusted to 1.5% in PBS was added. For the slide assay, 25 µL of bacteria suspended in 2% mannose was mixed with 25 µL of 1.5% RBC. Agglutination was visible as a ring of aggregated cells around a pellet of settled RBC in the 96-well assay or as clumping in the slide assay. Digital images of partial wells or 2 different slide fields were captured by photomicroscopy using the 4× lens of an EVOS M5000 Imaging System (Thermo Fisher Scientific, Waltham, MA, USA) after 24 h at 4 °C (96-well assay) or 1 h at 23 °C (slide assay). The controls included RBC only and H10407-P (plasmid-cured, CFA/I-deficient strain) [43] grown in the same manner as H10407.

Preparation of ETEC vaccines. The psoralen- and formalin-killed vaccines (PUVA–ETEC and formalin–ETEC, respectively) were prepared as described and stored in 25% glycerol-LB media at −80 °C. Vaccine aliquots were thawed, washed three times, and adjusted to either 2 × 10^9^ CFU equivalents/mL (OD_600_ 2.0) or 2 × 10^8^ CFU equivalents/mL (OD_600_ 0.2). An equal volume of PBS or dmLT adjuvant (kindly provided by Dr. Elizabeth Norton, Tulane University School of Medicine), to achieve a final concentration of 2.5 µg/mL, was added prior to the intramuscular (IM) delivery of 0.1 mL per mouse. This preparation yielded a dose of 1 × 10^8^ (high) or 1 × 10^7^ (low) inactivated ETEC ± 0.25 µg dmLT. 

Vaccination of mice with ETEC vaccines. Mouse vaccination experiments were conducted according to a protocol approved by the Wake Forest Institutional Animal Care and Use Committee (#A19-163). Female BALB/c mice (6–8 weeks old) were purchased from Charles River. Mice were housed in ventilated cages with unlimited water and commercially produced mouse chow. Mice (7 or 8 per group) were immunized with vaccines containing no adjuvant or combined with dmLT adjuvant at the indicated doses. The vaccination groups were: (1). PUVA–ETEC; (2). PUVA–ETEC + dmLT; (3). formalin–ETEC; and (4). formalin–ETEC + dmLT. The vaccines were administered intramuscularly in the hind flank on days 0, 21, and 42. Blood was collected on days 0 (pre-bleed), 34 (13 days post-boost 1), and 56 (14 days post-boost 2). Mice were monitored and weighed regularly after vaccination. Portions of sera from each group were pooled and stored along with individual samples at −80 °C.

ELISA. An ELISA assay using whole ETEC as the coating antigen was adapted from a published protocol [44]. ETEC cultured as per the inactivation experiments were harvested at 5 h, washed, and adjusted to an OD_600_ of 0.5 in sterile PBS. Microtiter plates (Nunc Maxisorp) were coated with the bacterial suspension (100 µL per well) and the plates were tightly sealed to prevent drying. After 18–24 h at 37 °C, the wells were washed three times with PBS and blocked with 200 µL/well of 5% Difco skim milk in wash buffer (PBS + 0.05% Tween20) for 2 h at 23 °C with agitation. After 3 washes, serum that had been serially diluted in PBS + 0.05% Tween20 + 1% skim milk (antibody buffer) was added (100 µL/well) and incubated for 2 h at 23 °C with agitation. The plates were washed 3 times and incubated with HRP-conjugated goat anti-mouse IgM+IgG+IgA (Millipore AP501P) or HRP-conjugated goat anti-mouse IgG (Invitrogen, Waltham, MA, USA) (100 µL/well), diluted 1:5000 in antibody buffer. After 1 h, the wells were washed and 100 µL of TMB substrate (3,3′, 5.5′-tetramethylbenzidine, #T0440, Sigma-Aldrich, Inc., St. Louis, MO, USA) was added. The plates were incubated in the dark for 20 min and the reaction was stopped by adding 100 µL of 2 M H_2_SO_4_. Absorption at 450 nm was measured using a BioTek EPOCH2NS spectrophotometer (BioTek Instruments, Inc, Highland Park, VT, USA). The serum antibody titer, defined as the dilution that yielded an OD_450_ value 4-fold greater than the assay background (antigen-coated wells with no serum/antibody added), was calculated using a linear regression analysis of the OD readings from the 3 to 4 Log_10_-transformed dilutions using the GraphPad Prism 9.0 software. The dilution series for the post-vaccination sera was started at 1:500, which generated no background signal in the uncoated (no antigen) control wells. Mouse anti-LPS (clone 2D7/1, #3564, Abcam, Waltham, MA, USA) or rabbit anti-*E.coli* (#4329-4906, BioRad, Hercules, CA, USA) paired with HRP-conjugated donkey-anti-rabbit IgG (#NA934OV, GE Healthcare, Chicago, Ill., USA) served as positive controls. To screen for ETEC protein-specific antibodies, ELISA plates were coated overnight at 4 °C with 100 µL/well of recombinant proteins, MipA, Skp, ETEC_2479, or EtpA (4 μg/mL in pH 9.6 carbonate buffer). The rest of the assay was as per the whole-cell ELISA.

LTB ELISA. The detection of LTB-specific antibodies was performed according to a published protocol [45]. Plates were coated with 100 µL of GM1 ganglioside diluted in PBS (0.5 µg/mL, Sigma-Aldrich #G7641) for 18–24 h at 23 °C. After 3 washes, the plates were blocked, and 100 μl of recombinant LTB (Sigma-Aldrich #E8656) diluted to 0.5 µg/mL in antibody buffer was added for 1 h at 23 °C. After 3 washes, 100 µL of mouse sera diluted serially in antibody buffer was added for 2 h at 23 °C. The rest of the protocol was as per the ETEC whole-cell ELISA. Rabbit anti-LTB, a gift from Dr. Eric Hall (Naval Medical Research Center, Bethesda, MD, USA), detected with anti-rabbit IgG-HRP secondary antibody, was used as a positive control. The results are expressed as OD450 after subtraction of the absorbance values from the wells that were identically treated but not coated with GM1. 

Statistics. Statistical significance was calculated by a two-tailed, unpaired Student’s *t*-test using the GraphPad Prism 9.0 software. *p* ≤ 0.05 was considered to be significant.

## 3. Results

### 3.1. Psoralen + UVA (PUVA) Inactivation of ETEC

To test the feasibility of psoralen-inactivated ETEC as a vaccine platform, dose response experiments were performed with AMT psoralen and UVA light (Figure 1). The ETEC strain H10407 was grown to stationary phase (OD_600_ 1.5) in 1.0 mL CFA broth cultures. AMT was added for 1 h, after which, the bacteria were removed from the culture and exposed to 1 J/cm^2^ of UVA light. The number of CFU remaining was measured by plating dilutions on LB agar. At 1 J/cm^2^ of UVA, the reduction in CFU/mL was AMT dose-dependent, but no AMT dose killed all of the bacteria in the culture, with residual CFU remaining in some experiments even at the highest AMT dose (Figure 1a). Doubling the dose of UVA to 2 J/cm^2^ (Figure 1b) sterilized the cultures at AMT doses of ≥ 10 µg/mL; no residual CFU were detected after plating the entire culture on LB agar broth (depicted as symbols plotted on the x axis) or after re-inoculating the entire culture into fresh LB broth, which did not yield an increase in OD_600_ relative to the starting culture over 72 h. Similar CFU/mL were detected in the live (untreated) and AMT only cultures (filled circle and star, respectively), indicating that AMT at the applied doses was not toxic in the absence of UVA light. Similarly, as shown in Figure 1c, while UVA alone at doses up to 4 J/cm^2^ did not kill ETEC (open symbols), UVA at 2 and 4 J/cm^2^ killed the cultures completely in combination with 50 µg/mL of AMT (closed symbols). Therefore, an AMT dose of ≥ 10 µg/mL combined with at least 2 J/cm^2^ of UVA (Figure 1b) yielded no residual CFU. 

For vaccine purposes, the inactivation method should reproducibly sterilize a culture. As shown in Figure 1d, AMT at 50 µg/mL, followed by irradiation at 2 J/cm^2^, reproducibly killed the ETEC cultures. In six independent experiments representing >1 × 10^10^ CFU, no CFU remained. The inactivation controls were heat-killed (1 h at 65 °C) and formalin-killed. To test their capacity to repair psoralen adducts, samples of bacteria from five independent PUVA-inactivated cultures were diluted into fresh media. Approximately 1 × 10^8^ inactivated bacteria yielded no growth (no change in optical density) over 72 h at 37 °C. These results strongly suggest that ETEC were not capable of reverting to replication competency after treatment with 50 µg/mL of AMT + 2 J/cm^2^ of UVA according to the described protocol and at a small scale.

### 3.2. Metabolic Activity of PUVA–ETEC

We next tested whether ETEC inactivation with PUVA would result in a killed but metabolically active (KBMA) state, as reported for other psoralen-inactivated bacteria [12]. As shown in Figure 2a, ETEC inactivated with 50 µg/mL of AMT and 2 J/cm^2^ of UVA were KBMA, as determined by an MTS assay. Although the level of dehydrogenase enzyme activity was decreased relative to the live bacteria and varied among the independent PUVA-inactivated cultures, MTS activity was consistently detected in the PUVA-inactivated cultures, but not in the heat- or formalin-inactivated cultures. Frozen and thawed PUVA–ETEC remained KBMA, indicating that the property was stable to the short-term frozen storage of vaccine stocks in glycerol-containing media (Figure 2b). A time course analysis was performed, which demonstrated that metabolically active bacteria were detectable in the cultures of PUVA-inactivated ETEC for up to 24 h (Figure 2c). Identical wells set up for plating confirmed that no CFU were present in the PUVA-, formalin-, and heat-inactivated samples at the time of MTS addition to the wells, indicating that the MTS activity was not a result of residual live bacteria. Taken together, these results demonstrate that treatment with 50 µg/mL of AMT followed by 2 J/cm^2^ of UVA light generated replication-incompetent ETEC that were KBMA. 

### 3.3. CFA/I-Mediated MRHA after PUVA and Formalin Inactivation of ETEC

We next asked if the inactivation method influenced surface protein structure/conformational state, using protein function as a read-out. The functional capacity of CFA/I fimbriae to agglutinate human Type A red blood cells (RBC) in the presence of mannose [42] (MRHA assay) was used as a surrogate measure of the protein structural/conformational integrity after inactivation with PUVA or formalin. Live or inactivated ETEC H10407 or H10407-P (plasmid cured, CFA/I-negative [46]) were incubated with RBC in the presence and absence of mannose using a 96-well round bottom plate and slide assays, as described in the methods. As shown in Figure 3a, live ETEC agglutinated the RBC in the presence of mannose, a property of CFA/I fimbriae. ETEC inactivated via the PUVA method (50 µg/mL of AMT + 2 J/cm^2^ of UVA light) agglutinated the RBC in a manner that was qualitatively indistinguishable from live bacteria. In contrast, a reduced capacity of formalin–ETEC to agglutinate RBC under the same experimental conditions was apparent. H10407-P failed to agglutinate RBC due to the absence of the gene encoding CFA/I. Similar results were obtained using the slide assay, with two fields captured using a 4× lens shown for each sample (Figure 3b). These results support the hypothesis that, in targeting nucleic acid, PUVA inactivation preserves a key antigenic surface protein target of the immune response in a native-like conformation, as demonstrated by the retention of CFA/I adhesive function for RBC, while formalin modified the protein in a manner that compromised this adhesion. 

### 3.4. Immunogenicity of PUVA-Inactivated ETEC (PUVA–ETEC) Vaccine in Mice

We next evaluated the vaccines’ immunogenicity in mice. In order to measure the humoral response generated against a broad repertoire of H10407 ETEC antigens after vaccination, an ELISA was developed using whole, untreated bacteria as the coating antigen, according to a published method [44]. To evaluate the consistency of the well coating, crystal violet staining was used after the application of whole ETEC to Nunc Maxisorp wells. Using washed CFA broth cultures adjusted to 0.5 OD_600_ units in sterile PBS, the crystal violet staining revealed consistent ETEC coating (mean OD_630 nm_, 0.103 ± 0.009, n = 72 wells). The ELISA assay was tested using broadly reactive rabbit anti-*E. coli* and mouse-anti-LPS antibodies. As shown in Figure 4a,b, a dose–response was generated against whole ETEC and no signal was generated in the uncoated wells (Figure 4a). 

As an initial test of its immunogenicity in mice, the PUVA–ETEC vaccine was administered with or without a dmLT adjuvant in an IM prime-boost regimen using a vaccine dose of 1 × 10^8^ CFU equivalents ± 0.25 µg dmLT. The dmLT dose was based on previous studies in mice that demonstrated an effective range of 0.10–1.0 µg of dmLT per vaccine dose [31]. BALB/c mice (5/group) were primed on day 0, boosted on day 21, and their total ETEC-specific serum immunoglobulin (IgM+G+A) was measured at 2 weeks post-boost. Figure 4c depicts the OD450 values for the pooled pre-immune and 2 weeks post-boost sera from the two vaccination groups. ETEC-specific antibodies were detected in the post-boost sera from both groups (open symbols), but not in the pre-immune sera (closed symbols). The addition of dmLT to the vaccine did not increase the antibody levels (open circles vs. open triangles). Therefore, in the context of the dosing and vaccination regimen used, the PUVA–ETEC vaccine was immunogenic in mice, but the antibody response measured (anti-ETEC serum Ig) was not enhanced by dmLT. 

### 3.5. Immunogenicity of PUVA–ETEC as Compared to Formalin–ETEC Vaccines in Mice

Next, the immunogenicity of the PUVA-inactivated ETEC vaccine was compared to a formalin-inactivated counterpart. For this experiment, the vaccine dose was reduced from 1 × 10^8^ CFU equivalents to 1 × 10^7^ CFU equivalents per mouse. The mice in both groups initially lost weight after receiving the 1 × 10^8^ dose (Figure 5a)**,** and one mouse in the dmLT group was sacrificed after excessive weight loss. This adverse reaction was likely to due to endotoxin. The weight loss was coincident with decreased activity, but the remaining mice recovered within 48 h considering the criteria of activity level and weight gain. No changes in fecal consistency or other body condition parameters were noted. No injection site reactivity was noted. We also considered the possibility that the high antigen dose may have masked any adjuvant effect of the dmLT, which has been reported to depend on the ratio of adjuvant to antigen, the nature of the antigen, and other factors [2,47]. 

The mice were vaccinated with low-dose vaccines ± 0.25 µg of dmLT in four groups: (1) PUVA–ETEC, (2) PUVA–ETEC + dmLT, (3) formalin–ETEC, and (4) formalin–ETEC + dmLT. In contrast to the high-dose vaccines, there was no adverse effect in terms of body weight change after vaccination with 1 × 10^7^ CFU equivalents (Figure 5b). Sera from the vaccinated mice collected at days 20 (post-prime), 34 (13 days post-boost), and 56 (14 days post-boost 2) were screened for ETEC-specific IgG (Figure 6). Both vaccines gave rise to serum IgG that increased by an order of magnitude after the first boost (day 20 as compared to day 13 post-boost) and only modestly after the second boost. The titers were significantly increased by the co-administration of dmLT, except for in the case of the formalin vaccine at day 20. However, no significant differences were found in the magnitude of the serum IgG response induced by the PUVA- and formalin-inactivated vaccines when measured against the whole ETEC bacteria as the coating antigen.

### 3.6. Reactivity of Immune Sera with Heterologous ETEC Strain

The PUVA–ETEC and formalin–ETEC vaccines were prepared from strain H10407 (078-H11). Serum pools from each vaccination group were next screened against ETEC B7a (O148:H28) [40], with their reactivity against the homologous strain H10407 measured in parallel. As shown in Figure 7, the IgG titers recognizing B7a were significantly reduced relative to H10407, but the difference was less than an order of magnitude, indicating a high cross-reactivity of the H01407-immune sera with the B7a strain. DmLT enhanced the IgG titers reactive against both strains (*p* < 0.01 for all ± dmLT comparisons). Therefore, both vaccines induced robust serum IgG antibodies that recognized a heterologous strain with a different colonization factor profile relative to the vaccine strain, presumably reflecting an abundance of shared protein and polysaccharide antigens. 

### 3.7. IgG Response Raised against Conserved ETEC Proteins

Having determined that the polyclonal anti-ETEC serum antibody response (representing a multitude of antigens) was similar for the two vaccines, we tested the reactivity of the immune sera against individual ETEC proteins. ETEC proteins that are conserved across clinical isolates were chosen for this analysis. The pooled sera were first screened against Mip, Skp, and ETEC_2479, which were identified in a screen for immunogenic outer membrane proteins that mediated protection against several ETEC strains in mice [38,48]. As shown in Figure 8a, IgG that recognized all three proteins was present at a low level in the pooled sera of the PUVA–ETEC-vaccinated mice (detected at a 1:300 dilution of serum) and in each case, the PUVA–ETEC vaccine raised a significantly greater antibody response, both with and without dmLT, than the formalin–ETEC vaccine did. The response against EtpA, which mediates the adhesion of ETEC to small intestinal epithelial cells and is a protective antigen in mice [49,50], was also measured. The IgG levels were again modest, in that they were detected only at 1:100 and 1:300 serum dilutions. Again, the PUVA–ETEC vaccine induced significantly greater IgG against EtpA in the presence and absence of dmLT than the formalin–ETEC vaccine did (Figure 8b). DmLT improved these IgG levels when combined with the PUVA vaccine (*p* = 0.002 and *p* = 0.005 for the 1:100 and 1:300 serum dilutions, respectively). Mouse anti-EtpA antibody is shown in Figure 8c as a positive control. Together, the results indicate that the PUVA–ETEC vaccine preferentially gave rise to antibodies that recognized conserved ETEC proteins, supporting the hypothesis that PUVA inactivation is superior to formalin for maintaining the integrity of protein antigens presented in the context of a whole-cell ETEC vaccine. 

### 3.8. IgG Response Raised against Heat-Labile Toxin B Subunit (LTB)

Given the KBMA status of the psoralen-inactivated ETEC, we hypothesized that the PUVA–ETEC vaccine might induce LT-specific antibodies by virtue of the secretion or presentation of surface-associated LT [51] to the immune system. To test this hypothesis, an LT ELISA was performed using the pooled sera collected after the first and second boosts (Figure 9). Notably, LTB-specific antibodies were detected in the sera of the mice vaccinated and boosted once and twice with low-dose PUVA–ETEC, but not with formalin–ETEC (Figure 9a,c). As expected, anti-LT antibodies were present at a high level when dmLT was delivered with the PUVA- and formalin-killed vaccines, reflecting the intrinsic antigenicity of dmLT (Figure 9b,d) [30]. Next, sera from the high-dose vaccination experiment (Figure 4) were subject to the same assay. As shown in Figure 9e, mice that received the high-dose vaccine produced anti-LTB antibodies in the absence of dmLT, and again produced a significantly greater response when dmLT was co-administered with the vaccine. Prebleed sera did not react with LT. These results further distinguish the psoralen- from the formalin-inactivated vaccine, in that the psoralen inactivation method promoted the presentation of LT, a key secreted and cell-associated protein target of the immune response. 

## 4. Discussion

In the current study, we explored the potential of psoralen inactivation as an alternative to formalin to produce an inactivated vaccine for ETEC, for which whole-cell vaccine platforms provide a safe and relatively inexpensive means for targeting a large repertoire of antigens, including proteins conserved across clinical strains [52,53]. We determined the experimental parameters for the reproducible inactivation of ETEC with psoralen + UVA light (PUVA), and then compared the properties and immunogenicity of PUVA–ETEC to that of formalin-inactivated ETEC. 

We found that several properties of PUVA–ETEC were not shared by formalin–ETEC. First, PUVA–ETEC were killed but metabolically active (KBMA), as previously reported for other psoralen-inactivated Gram-negative and Gram-positive bacteria [12,19,20,22,23,54]. Second, PUVA–ETEC, but not formalin–ETEC, retained their functional CFA/I adhesive capacity, suggesting that the surface fimbrial protein integrity was preserved by PUVA. Third, in mice, IM prime/boost with PUVA–ETEC and formalin-ETEC induced similar ETEC-specific serum IgG titers when screened against whole bacteria; however, when screened against specific ETEC proteins, PUVA–ETEC induced a greater IgG response against conserved ETEC proteins. Finally, unlike formalin–ETEC, PUVA–ETEC gave rise to IgG specific for heat-labile toxin (LT) in the absence of dmLT adjuvant. Together, the data support PUVA as a promising alternative to formalin to improve the immunogenicity of whole-cell ETEC vaccines. 

As reported previously for other bacteria [12,19,20,22,23,54], PUVA generated replication-defective ETEC that maintained its metabolic activity (KBMA), which is not a property of formalin-inactivated bacteria. Previous work described the KBMA state in psoralen-inactivated bacteria with mutations introduced at the *uvrABC* locus, which renders bacteria highly susceptible to UV-induced DNA damage by disabling the nucleotide excision repair system (NER) [19,55]. In our study, we achieved the complete and irreversible inactivation of wild-type ETEC with 50 µg/mL of AMT psoralen + 2 J/cm^2^ of UVA light. To our knowledge, we are the first to use AMT to photochemically inactivate bacteria. Because our goal was the complete and irreversible inactivation of bacterial replication, we used AMT doses that were 10–100 fold higher on a molar basis than those reported for other psoralen drugs used to inactivate bacteria for the purpose of vaccine production [20,23,54]. Since the frequency of adducts introduced into the genome is psoralen dose-dependent [19], the high AMT dose likely resulted in damage that exceeded the cellular repair capabilities. Also of note in this regard is a report that wild-type *E. coli* were inefficient at repairing inter-strand crosslinks induced by a different psoralen drug, 8-methoxypsoralen [56]. Nevertheless, generating PUVA–ETEC vaccine strains in a *uvr* mutant background remains an option for minimizing the potential for reversion to replication competency, which would present a significant safety concern.

A key outcome of our study is evidence to support the hypothesis that PUVA preserves proteins in native-like form, which we determined using protein adhesive function as a proxy for protein structural and antigenic integrity. An MRHA assay demonstrated that PUVA–ETEC retained its CFA/I-mediated adhesive function, which was indistinguishable from that of live ETEC, while the activity of formalin–ETEC was severely reduced, strongly suggesting that PUVA preserved while formalin compromised the CFA/I protein integrity. While it is clear that anti-CFA/I antibody responses are generated in animals and humans in the context of formalin-inactivated ETEC vaccines [32,41], and subtle negative effects of formalin on CFA/I antigenicity have been noted [57], the functional adhesion assay performed here is likely a more sensitive read-out of changes to CFA/I protein integrity. The question of protein antigenic epitope integrity after PUVA vs. formalin treatment of CFA/I-expressing ETEC strains could be addressed directly by measuring the reactivity of the monoclonal antibodies specific for the linear and conformational epitopes of CFA/I (and other outer membrane proteins), as per studies with Dengue virus, Rotavirus, and *Shigella* [2,14,16]. 

To compare the immunogenicity in mice, we chose an IM prime/boost regimen. Although there was no difference between the anti-ETEC serum antibody response measured by screening immune sera against whole bacteria, we considered that any reduction in antibody responses against individual proteins would be masked in the context of screening against the large repertoire of antigens displayed by the whole bacteria. When the sera were screened against the individual proteins that were chosen for conservation among the ETEC clinical isolates, IgG specific for Mip, Skp, ETEC_2479, and EtpA was detected after PUVA–ETEC vaccination, but the IgG levels were minimal after vaccination with formalin–ETEC. Previous studies with those recombinant proteins delivered to mice have demonstrated the induction of functional (adhesion blocking) antibodies and protection against a challenge with ETEC, including against heterologous strains [38,48,58], indicating that the recombinant proteins present in the vaccines resembled those presented in native form by infectious bacteria. We interpret our results to mean that those four conserved proteins were faithfully presented by the PUVA–ETEC vaccine in a conformation that closely resembled their native form, while the same proteins presented by the formalin–ETEC vaccine were altered in a way that compromised their antigenicity. We acknowledge that we tested one formalin dose and protocol for ETEC inactivation based on a published ETEC vaccine study [41]. Although the magnitude of the observed differences between the PUVA and formalin–ETEC described here could be a function of the dose/time/temperature chosen for the formalin inactivation, as shown in a study with *Shigella* [2], the results are nevertheless consistent with PUVA treatment preserving immunogenicity by sparing proteins from modification. 

The immune sera from the PUVA– and formalin–ETEC-vaccinated mice had high titers of IgG that recognized the vaccine strain (H10407) as well as B7a, a strain with a different serotype and distinct set of adhesins and colonization factors [59]. It is well documented that an effective humoral response against ETEC relies on antibodies directed against protein antigens, including colonization factors, coli surface proteins, other adhesins, an expanding group of proteins that are conserved among clinical isolates, and heat-labile toxin (LT) [36,37]. Given that antibodies directed against conserved surface-associated proteins were detected (Figure 8), the high level of cross-reactivity that we observed with B7a likely reflects the IgG generated against shared proteins, as well as shared polysaccharide antigens, including non-serotype-specific epitopes associated with LPS.

Although we demonstrated that the KBMA state is a property of psoralen-inactivated ETEC, we did not formally address the implications of sustained metabolic activity for an ETEC vaccine. There are obvious implications, including the potential for toxin production to cause adverse effects, which could be addressed using strains with mutations that detoxify but leave the intrinsic antigenicity and adjuvant activity of LT intact [30,31]. Interestingly, we found that PUVA–ETEC induced LT-specific IgG in the absence of dmLT adjuvant, which was not a property of formalin–ETEC (Figure 9). The presentation of LT by PUVA–ETEC may have resulted from toxin secretion, or alternatively, it could reflect the presentation of LPS-associated LT [51] in an immunogenic conformation that was not retained after formalin treatment. Either way, the finding implies a preferential presentation of LT, a key protein target of the protective immune response against ETEC infection by the psoralen-inactivated vaccine. The extent of active LT production by PUVA–ETEC should be examined in light of these data. A role for the KBMA state in promoting vaccine immunogenicity was shown for psoralen-killed *Bacillus anthracis*, which presented protective antigen (PA), a secreted protein, to the immune system in animal models [23]. Similarly, psoralen-inactivated *Listeria monocytogenes* produced listeriolysin O, promoting the cytosolic delivery of a heterologous antigen in a mouse model [19]. The presentation of LT in our study did not demonstrate that KBMA was consequential for the PUVA–ETEC vaccine, but the role of KBMA in immunogenicity could be tested by vaccinating mice with PUVA-inactivated H10407 constitutively overexpressing an immunogenic secreted protein such as EatA [60] and measuring the anti-EatA antibody response. 

Our results with dmLT recapitulate the results of studies demonstrating an adjuvant effect that was antigen dose-dependent (Figure 4 and Figure 6). DmLT (0.25 µg) delivered IM with 1 × 10^7^ CFU, but not 1 × 10^8^ CFU, equivalents of PUVA–ETEC via the intramuscular-route-augmented ETEC-specific serum IgG titers. Similar studies with *Campylobacter* and *Shigella* vaccines demonstrated the antigen dose-dependence of dmLT adjuvant activity [2,61]. We chose a parenteral route to test the immunogenicity of the ETEC vaccines and found that 1 × 10^8^ bacteria, with and without dmLT, caused weight loss, which was not observed at the 1 × 10^7^ dose (Figure 5). Although it is reasonable to propose that endotoxin caused the adverse effect, we cannot rule out an effect of the sustained production of LT by KBMA ETEC. This question could be addressed in part by comparing a high-dose formalin–ETEC vaccine (which does not produce LT) to high-dose PUVA vaccine. For the sole purpose of the initial comparisons of immunogenicity, we administered vaccines via the IM route. However, since whole Gram-negative bacterial vaccines are not practical for parenteral delivery due to their potential for adverse reaction due to endotoxin, future studies in mice should test physiologically relevant oral vaccination with PUVA-ETEC + dmLT, and include an evaluation of mucosal IgA responses, which would more closely model human vaccinations with ETVAX [32]. 

In summary, the current study fills a gap in the knowledge by directly comparing the properties and immunogenicity of psoralen-inactivated bacteria, first described more than a decade ago for their potential to improve whole-cell bacterial vaccines [12,19], to formalin inactivation, an industry standard. Using ETEC as a model, we presented evidence that psoralen inactivation expanded the repertoire of bacterial antigens presented to the immune system. The results provide a foundation for future studies to test the two vaccines against each other in murine challenge models. A proof of concept for the potential of the PUVA method to improve whole-cell, inactivated ETEC vaccines could be achieved with the ETVAX CS6-overexpressing strain, in which the CS6 antigen was negatively impacted by formalin [62]. Mouse studies supporting the generation of protective immune responses, as per studies with psoralen- versus formalin-killed Dengue virus vaccine candidates [16,18], would justify pursuing this approach for ETEC. However, significant challenges to advancing psoralen-killed bacterial vaccines remain. The methodology is at an early exploratory stage. The scalability for the production of large volumes of vaccine and the stability of protein function/antigenicity after psoralen inactivation need to be addressed. For the PUVA–ETEC vaccine described here (wild-type toxin and nucleotide excision repair genes), safety/toxicity testing in mice is essential, including assays for toxin production and replication-competent bacteria (residual or revertants) in vaccine preparations and tissues. With these significant challenges considered, our results imply that, by preserving the antigenic structure of proteins, including those that are conserved and promote protective immune responses, an ETEC vaccine inactivated with psoralen represents a potential improvement over existing inactivated vaccines. In addition, the psoralen approach could add value to multivalent vaccine approaches that use inactivated bacteria as a platform and/or to enhance the safety of live, attenuated vaccines [53]. 

## Figures and Tables

**Figure 1 microorganisms-11-02040-f001:**
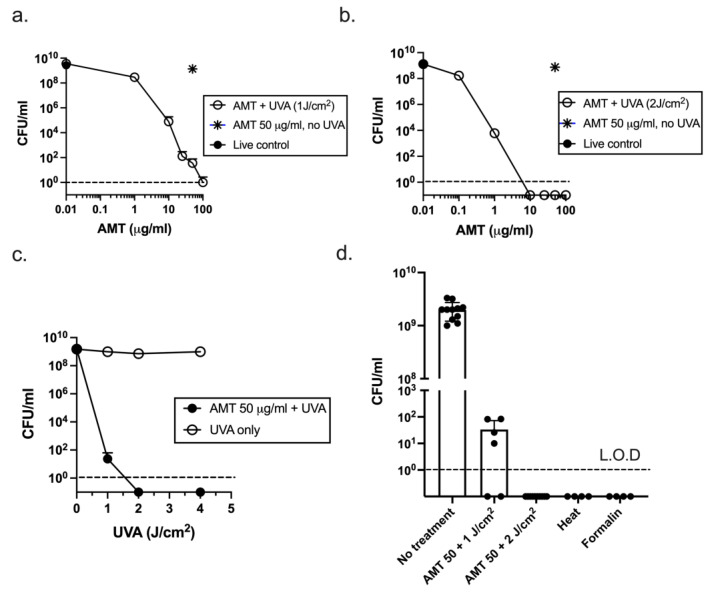
Inactivation of ETEC by Psoralen + UVA (PUVA). AMT psoralen at the indicated concentrations was added to 5 h CFA broth cultures for 1 h followed by irradiation with UVA light at (**a**)**.** 1 J/cm^2^ and (**b**). 2 J/cm^2^; (**c**). AMT psoralen at 50 μg/mL was added to cultures followed by irradiation at 1, 2, and 4 J/cm^2^. Results depict CFU/mL remaining (mean ± s.d.) from 2–3 experiments/dose; L.O.D = limit of detection; (**d**). Reproducibility of killing method; independent experiments in which 1.0 mL stationary phase ETEC cultures were treated with 50 µg/mL AMT and 1 J/cm^2^ (n = 6) or 2 J/cm^2^ UVA (n = 6); 0 CFU is plotted on the x axis. Inactivation controls: heat (n = 4), formalin (n = 4); mean CFU/mL ± s.d. is shown.

**Figure 2 microorganisms-11-02040-f002:**
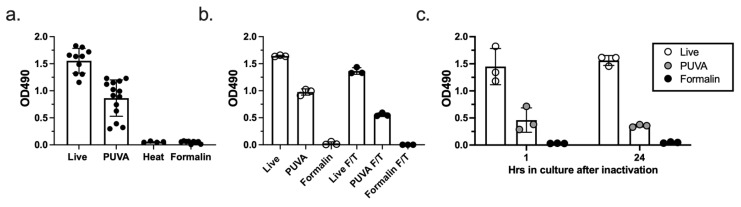
PUVA-ETEC are KBMA. (**a**). MTS assay for metabolic activity. Bacteria treated with 50 µg/mL AMT + 2 J/cm^2^ UVA were washed and added to 96-well cultures with MTS reagent for 1 h. Bars depict the mean OD490 ± s.d. from multiple independent cultures (dots). (**b**). MTS assay on frozen and thawed bacteria, (n = 3 per sample); (**c**). KBMA activity over time. Live, PUVA-ETEC, and formalin-ETEC were added to fresh 96-well cultures. Inactivated bacterial densities were adjusted to the 24 h OD600 of live bacteria to normalize for their inability to replicate over 24 h. At 1 h and 23 h after initiation of cultures, MTS reagent was added for an additional 1 h and the absorbance at OD490 nm was measured; the mean OD490 ± s.d. from 3 cultures is shown.

**Figure 3 microorganisms-11-02040-f003:**
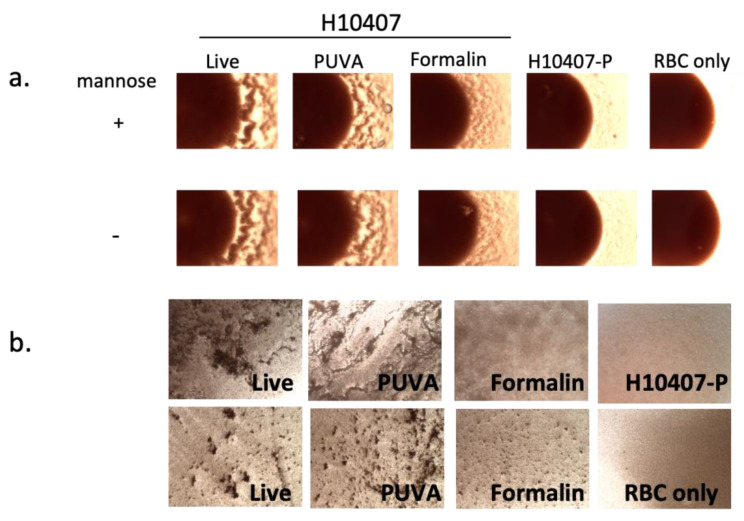
MRHA activity of live, PUVA- and formalin-inactivated ETEC. Bacteria were mixed with (**a**). human Type A RBC ± mannose in a 96-well round bottom plate or (**b**). with human Type A RBC + mannose on glass slides as described in methods. Photomicrographs captured under visible light with an RGB filter at 4× magnification using an EVOS M5000 Imaging System depict (**a**). partial wells showing agglutination visible at RBC pellet edges after 24 h of settling or (**b**). agglutination visible as clumping on slides after 1 h; 2 fields are shown for each sample in (**b**). Controls were RBC only and H10407-P [43], which lacks the pCS1 virulence plasmid encoding CFA/I [46]. Well and slide assays were performed at least 3 times with similar results.

**Figure 4 microorganisms-11-02040-f004:**
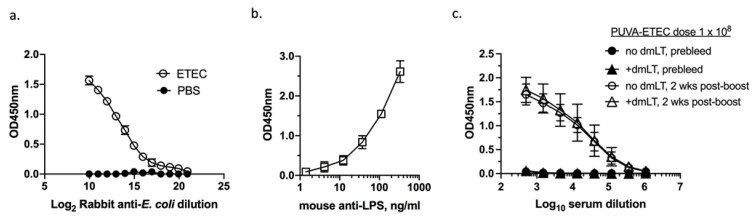
ETEC whole cell ELISA and immunogenicity of high dose PUVA–ETEC vaccine in mice. Nunc Maxisorp wells were coated with live H10407 ETEC (OD_600_ = 0.5) suspended in PBS; no antigen control wells received PBS; (**a**). ETEC-coated or PBS control wells probed with rabbit anti-*E.coli* antiserum; (**b**). ETEC-coated wells probed with mouse anti-LPS antibody; (**c**). serum antibody response (IgM+G+A) of BALB/c mice primed and boosted with 1 × 10^8^ CFU equivalents of PUVA–ETEC (n = 5) or PUVA–ETEC + dmLT (n = 4). Pooled sera collected on day 0 (prebleed) and day 35 (2 weeks post-boost) were analyzed. Mean OD_450_ ± s.d. of triplicate pools is shown in each panel. One of five mice in the PUVA–ETEC + dmLT group (panel (**c**)) was sacrificed before the end of the experiment due to excessive weight loss.

**Figure 5 microorganisms-11-02040-f005:**
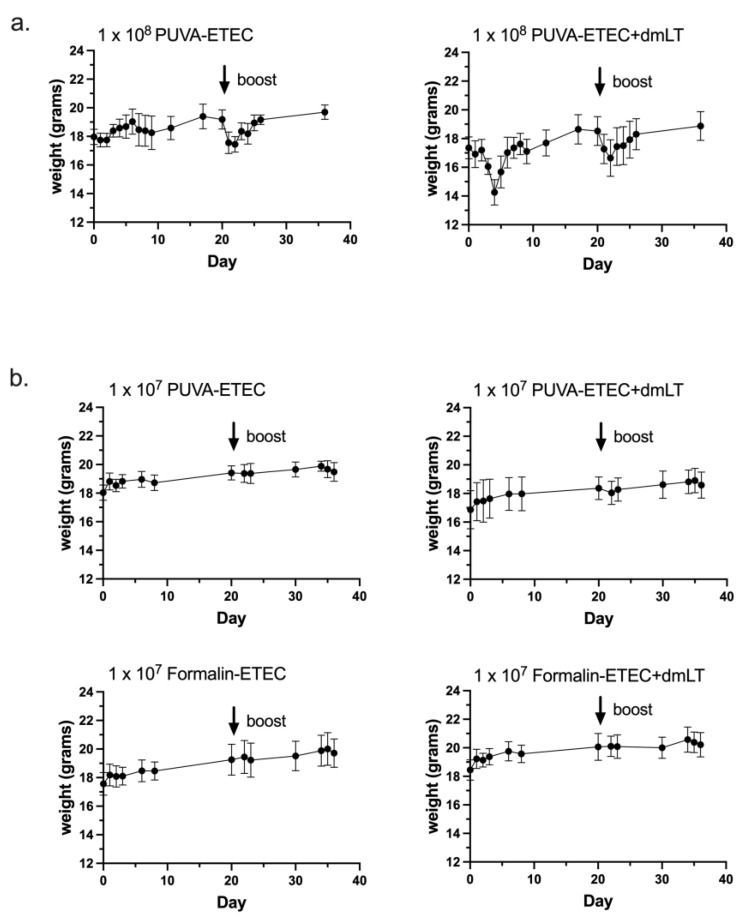
Body weight change curves after vaccination. Balb/c mice were weighed before intramuscular vaccination (time 0), and then at time points post-prime and post-boost with (**a**). high-dose PUVA–ETEC ± dmLT (1 × 10^8^ CFU equivalents + 0.25 µg dmLT); 1/5 mice in the high dose + dmLT group was sacrificed due to excessive weight loss; (**b**). low-dose PUVA–ETEC or formalin–ETEC ± dmLT (1 × 10^7^ CFU equivalents + 0.25 µg dmLT); points represent mean ± s.d. from 7 or 8 mice per group.

**Figure 6 microorganisms-11-02040-f006:**
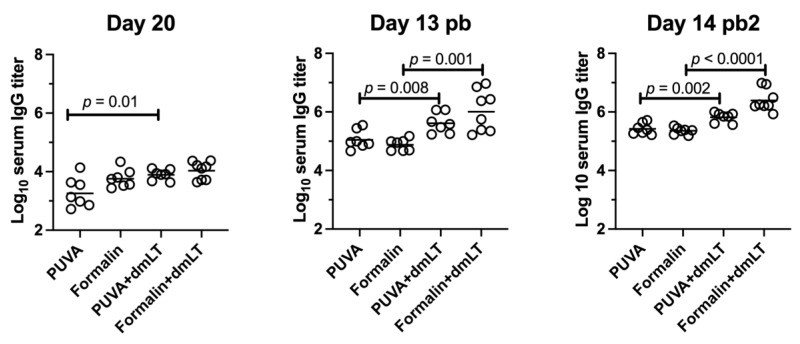
Immunogenicity of low-dose PUVA–ETEC as compared to formalin–ETEC in mice. BALB/c mice (n = 7 or 8) were vaccinated on Day 0 and boosted on Days 21 and 42 with 1 × 10^7^ CFU equivalents vaccines ± dmLT. Sera were analyzed for ETEC-specific IgG on day 20 after the priming dose, on day 13 after the 1st boost (Day 13 pb), and on day 14 after the 2nd boost (day 14 pb2) using ETEC H10407 as the coating antigen. Titer was defined as the dilution of serum that yielded a OD450 signal 4-fold above background as described in methods. The geometric mean titer (GMT) is indicated by the bar. Significance was determined by unpaired Student’s *t* test.

**Figure 7 microorganisms-11-02040-f007:**
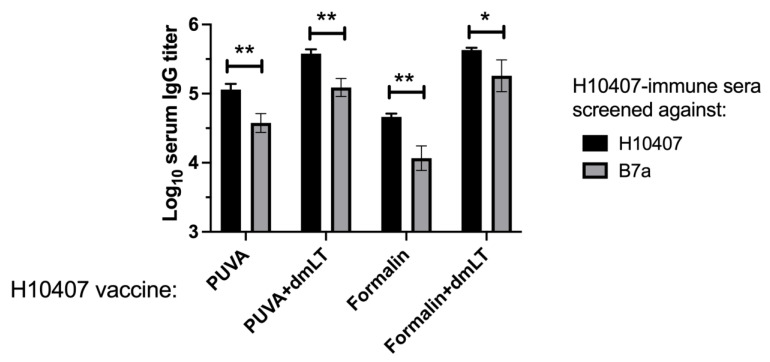
Reactivity of H10407-immune serum against ETEC strain B7a. Pooled sera collected on day 13 post-boost were screened using whole B7a or H10407 bacteria as the ELISA coating antigen. Data depicted in the figure represent 3 independent pools screened against H10407 and B7a. *** p* < 0.01; ** p* = 0.05, unpaired Student’s *t*-test.

**Figure 8 microorganisms-11-02040-f008:**
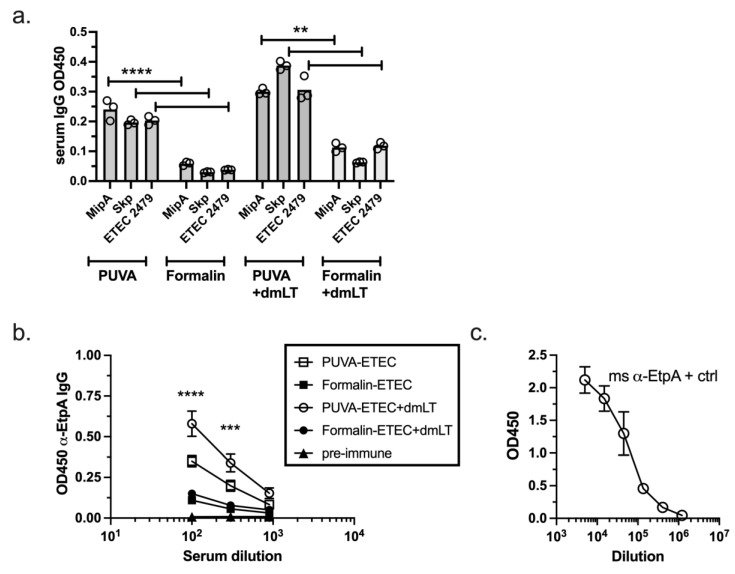
Vaccine-induced responses to conserved ETEC proteins. (**a**). Pooled sera collected day 13 post-boost screened against Mip, Skp, and ETEC 2479 proteins at 1:300 dilution. The bar represents the mean OD_450_ ± of triplicate samples from 1 of 3 experiments with similar results. **** *p* < 0.0001, PUVA vs. formalin for each protein; *** p* < 0.01, PUVA + dmLT vs. formalin + dmLT for each protein. (**b**). Pooled sera from day 14 post-boost 2 screened against EtpA. Data are the mean ± s.d. of OD450 values of 2 independent experiments. **** *p <* 0.0001 and *** *p =* 0.0002, PUVA vs. formalin at 1:100 and 1:300 serum dilutions. (**c**). polyclonal mouse anti-EtpA antibody titration against 0.4 µg EtpA per well.

**Figure 9 microorganisms-11-02040-f009:**
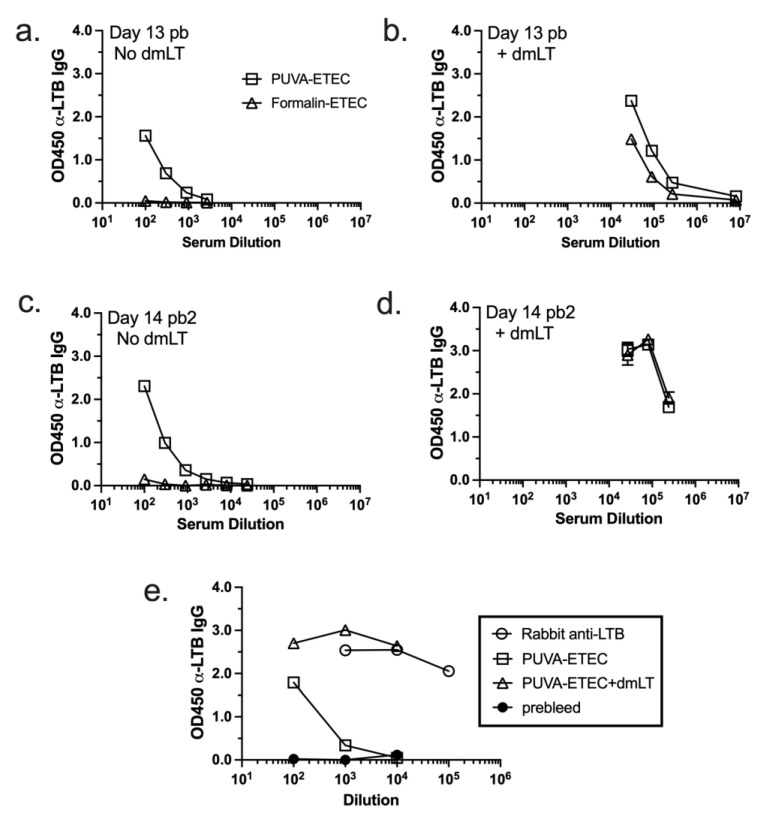
Vaccine-induced responses to heat labile toxin subunit B (LTB). Pooled sera collected on day 13 after the 1st boost (day 13 pb, (**a**,**b**)) and on day 14 after the 2nd boost (day 14 pb2, (**c**,**d**)) from mice that received low dose vaccines were screened for reactivity against LTB. (**e**). Pooled sera collected 2 weeks after boost with high dose PUVA vaccine ± dmLT. Rabbit anti-LTB served as a positive control. The mean OD_450_ ± sd from duplicate or triplicate ELISA values are shown.

## Data Availability

Data are contained within the article.

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
