# Peer review of "The Immunogenicity and Properties of a Whole-Cell ETEC Vaccine Inactivated with Psoralen and UVA Light in Comparison to Formalin"

_microorganisms, 2023, doi:10.3390/microorganisms11082040_

Round 1

Reviewer 1 Report

This is a very interesting and well-presented study in an important area viz. the impact of retaining metabolic activity and protein structure in ETEC vaccines. The study is presented in a logical and clear manner - and moves in a step-by-step manner from assessing dose responses, retention of bacterial function (binding) and metabolic activity, to vaccine preparation and monitoring of immunogenicity in mice.

The key findings are important - and support the hypothesis the PUVA affords the potential to retain immunogenicity to conserved ETEC proteins. The finding that heterologous bacterial responsiveness was observed (i.e. vaccine induce antibody cross reactivity) was also interesting.

The Discussion does allude to the major concern of this study viz. retention of metabolic activity and the danger of continued toxin production. This should have been tested. There is no indication of the impact of PUVA vaccination on general health of mice - except for one having to be sacrificed due to weight loss. The high dose vaccinated mice did show weight loss at vaccination and after boosting, but this was not apparent in low dose groups (supplementary data). Were any sera retained and could they be checked for residual toxin? Were dissections performed on any mice? Did vaccination impact on their eating or bowel movements? This is a very major issue as safety is paramount in proposing new vaccine approaches. If any residual toxin production is detectable and/or symptoms of diarrheal disease or general ill health - this would require a reassessment. The weighing up of better immunogenicity vs. risk is paramount.

I think the paper would benefit with more detail about the mice experiments (general health and any other measures performed). As it stands it presents an important body of work (despite the major caveat).

On page 2 line 92 - E. coli rather than just coli. Figure 4c - it appears that the +dmLT prebleed data (closed triangles) tracks with the post-boost data - which doesn't make sense. Either the figure is unclear - or needs explanation, as one should not have a titer in pre-bleed samples?

Author Response

  1. potential danger of retention of metabolic activity and continued toxin production: Response: Valid point given that our immunological data demonstrate that toxin was produced by psoralen-inactivated bacteria, at least at a level sufficient to give rise to anti-LT serum IgG. Since psoralen-inactivated bacteria do not replicate, we would not expect to be able to detect LT in serum at the time points that blood was collected for antibody testing (20 days post vaccination or 14 days post-boost).  A sentence in the discussion remains proposing the option of producing vaccine strains in a toxin mutant background (lines 529-531). Agreed that more discussion of immunogenicity vs risk of KBMA ETEC is needed. Re: general health of vaccinated mice, we have included the weight data as a primary figure instead of a supplementary figure since the point is worth emphasizing (now Fig. 5). The text in the results section is now expanded to address general health of the mice at low and high dose (lines 370-373). The results text describing the low dose vaccine was moved to lines 388-399, placing it in between the new weight figure (Fig 5) and the low dose vaccine immunogenicity figure (now Fig. 6). Subsequent figure numbers are corrected throughout.  Challenges related to KBMA and toxin are expanded in the discussion: lines 591-592, 609-612, 635-638.

  1. Page 2 line 92 of original: E.coli rather than just coli: Response: correct as originally written- “coli surface proteins” (line 93 in revised)

  1. Figure 4c clarification: the +dmLT prebleed data appeared to “track with post-boost data.” Response: The figure is correct; the +dmLT prebleed line (closed triangle) is at baseline (no titer) on the graph. Difficulty may be that the prebleed lines overlap (closed symbols) as do the post-boost lines (open symbols). In the results text, the symbols are mentioned with the description of the data (lines 345-6).  To clarify further, the data order in the legend was changed to start with prebleeds and follow with post-boost, and the prebleed symbols were increased in size to facilitate viewing of the 2 overlapping lines. Modified Fig 4c inserted above the original figure.

Reviewer 2 Report

This is a well-written paper demonstrating a positive new vaccine approach. Just a few minor concerns.   More description is needed of the dmLT adjuvant in the introduction.  What does it consist of?  One example was given of its prior use, but has it been used in vaccines other than the one listed?

L24 define dmLT

Why haven't psoralen vaccines been adopted? Due to the risk of toxin production from KMBA cells?  It is a recent line of study?  Another sentence or two of discussion is needed for this point as otherwise it seems like there must be a glaringly negative factor that is being deliberately avoided.

L405 Please define OMPs

L487 Was possibility of repair of psoralen damage as mentioned here checked?  How would a study to determine this be designed? Even if this repair is highly inefficient, it would be disastrous is KMBA converted to live functional cells.

L570-582 Need to add a sentence about some potential problems such as toxin production to summary/concluding paragraph.  Even though the vaccine methodology described looks very promising, it is not without a few concerns. 

Author Response

  1. Insufficient information on dmLT adjuvant: Response: additional text describing dmLT has been added to the Introduction (lines 98-103) and dmLT was defined on line 24

  1. Why haven’t psoralen vaccines been adopted; is this a recent line of study? Response: this is an obvious and important question. Gaps in knowledge and challenges associated with the psoralen approach to bacterial vaccines were not sufficiently addressed and are now expanded in the discussion (lines 619-622 and 630-639).

  1. Define OMP: Response: OMP replaced with outer membrane proteins (line 445).

  1. Possibility of repair of psoralen damage: Response: this is an important point- further data in support of irreversible inactivation at small scale was added to the text in the results section (lines 254-259) and the potential for reversion to replication competent and implication for safety is expanded in the discussion (lines 529-531 and lines 635-639).

  1. Reference to potential concerns with psoralen vaccines needed in summary paragraph of discussion: Response: we have included  gaps and challenges with regard to psoralen-killed ETEC and bacterial vaccines in general in the summary paragraph (lines 630-639).